# Fast Recombination of Charge-Transfer State in Organic Photovoltaic Composite of P3HT and Semiconducting Carbon Nanotubes Is the Reason for Its Poor Photovoltaic Performance

**DOI:** 10.3390/ijms24044098

**Published:** 2023-02-17

**Authors:** Mikhail N. Uvarov, Elena S. Kobeleva, Konstantin M. Degtyarenko, Vladimir A. Zinovyev, Alexander A. Popov, Evgeny A. Mostovich, Leonid V. Kulik

**Affiliations:** 1Voevodsky Institute of Chemical Kinetics and Combustion of the Siberian Branch of the Russian Academy of Sciences, 630090 Novosibirsk, Russia; 2Siberian Physico-Technical Institute of the Tomsk State University, 634050 Tomsk, Russia; 3Rzhanov Institute of Semiconductor Physics of the Siberian Branch of the Russian Academy of Sciences, 630090 Novosibirsk, Russia; 4Laboratory of Organic Optoelectronics of the Novosibirsk State University, 630090 Novosibirsk, Russia

**Keywords:** organic photovoltaics, semiconducting polymer, carbon nanotube, EPR, charge-transfer state

## Abstract

Although the photovoltaic performance of the composite of poly-3-hexylthiophene (P3HT) with semiconducting single-walled carbon nanotubes (s-SWCNT) is promising, the short-circuit current density *j_SC_* is much lower than that for typical polymer/fullerene composites. Out-of-phase electron spin echo (ESE) technique with laser excitation of the P3HT/s-SWCNT composite was used to clarify the origin of the poor photogeneration of free charges. The appearance of out-of-phase ESE signal is a solid proof that the charge-transfer state of P3HT^+^/s-SWCNT^−^ is formed upon photoexcitation and the electron spins of P3HT^+^ and s-SWCNT^−^ are correlated. No out-of-phase ESE signal was detected in the same experiment with pristine P3HT film. The out-of-phase ESE envelope modulation trace for P3HT/s-SWCNT composite was close to that for the polymer/fullerene photovoltaic composite PCDTBT/PC_70_BM, which implies a similar distance of initial charge separation in the range 2–4 nm. However, out-of-phase ESE signal decay with delay after laser flash increase for P3HT/s-SWCNT composite was much faster, with a characteristic time of 10 µs at 30 K. This points to the higher geminate recombination rate for the P3HT/s-SWCNT composite, which may be one of the reasons for the relatively poor photovoltaic performance of this system.

## 1. Introduction

Single-walled carbon nanotubes (SWCNTs) are an interesting alternative to fullerenes or small molecules as the electron acceptor material for the active layer of organic photovoltaic (OPV) devices due to the high current mobility and large aspect ratio [1,2,3,4,5]. Promising results were obtained some years ago for the bulk heterojunction composite of poly-3-hexylthiophene (P3HT) and semiconducting SWCNTs (s-SWCNTs), with s-SWCNTs separated from a statistical mixture of metallic SWCNTs and s-SWCNTs [6]. However, paramagnetic intermediates of photoelectric conversion (either free photogenerated charges or charge-transfer states) were not detected for such composites. Thus, the mechanism of photoelectric conversion in such polymer/s-SWCNT systems is unclear, which complicates their further optimization. Although very high open-circuit voltage *V_OC_* was reported for OPV devices based on the P3HT/s-SWCNT composite, the short-circuit current density *j_SC_* (about 2 mA/cm^2^) [6] was significantly lower than the typical values for polymer/fullerene systems (more than 10 mA/cm^2^).

In order to find the origin of the relatively poor photovoltaic performance of P3HT/s-SWCNT composite, we studied the light-induced charge-transfer state (CTS) formed at the P3HT/s-SWCNT interface upon the exciton splitting. CTS is known as the principal intermediate of photoelectric conversion in organic photovoltaics and influences all parameters determining the performance of organic solar cells [7,8,9,10]. The experimental study of CTS is difficult because of its transient nature. The optical absorption and luminescence of CTS are often weak and masked by the corresponding signals from the other species (singlet and triple excitons, free charges) in the donor/acceptor composite. In favorable cases, some information about spectral and kinetic parameters of CTS can be obtained by the optical methods [11,12,13]. However, these methods are generally unable to provide information about the structure of CTS. This can be conducted routinely using a special version of electron spin echo (ESE) spectroscopy—out-of-phase (OOP) ESE. Here, “out-of-phase” means that the transversal component of the magnetization in the rotating frame is orthogonal to that of normal in-phase ESE produced by paramagnetic species in thermal equilibrium. The OOP ESE appears solely due to correlation between spin of the electron and the hole constituting the CTS. This correlation arises because the CTS precursor (the exciton formed upon light absorption by the donor or acceptor part of the composite) has a definite electron spin, usually zero (the singlet exciton). The triplet excitons can also contribute to the charge photogeneration in special cases [14]. An attractive feature of OOP ESE is that it is free from the contribution of any transient species other than CTS. Stable paramagnetic species also do not contribute to OOP ESE because they produce only in-phase ESE.

Historically, OOP ESE was observed for the first time when phase-sensitive ESE detection was applied to study light-induced processes of charge separation in the photosynthetic bacterial reaction centers (RCs) [15]. At that time, it was erroneously interpreted as arising from sequential electron transfer occurring in the course of the microwave pulse sequence. Later, the correct interpretation of this signal was put forward [16]. This took into account that the spin-correlated radical pair (SCRP) is generated upon fast light-induced electron transfer in the RC and that the electron spins of both radicals are excited by the strong microwave pulses. In this respect, OOP ESE can be viewed as arising due to the instantaneous diffusion [17] in the non-equilibrium system of two spins that are correlated initially. Later, OOP ESE was successfully used for determining the length of initial photo-induced electron transfer in bacterial photosynthetic reaction centers and plant photosystems [18,19,20,21,22,23,24,25,26] and also in model donor–acceptor compounds [27]. For donor–acceptor systems with well-defined geometry, the OOP ESE technique allows the determination of the distance between the positive and the negative charges within the geminate pair (another name of CTS) with angstrom precision [28,29]. Nowadays, it has become a convenient tool for the determination of the structure of CTS in various organic photovoltaic bulk heterojunction composites [30,31,32].

For the successful measurement of the distance between the components of SCPR using the OOP ESE technique, the following are prerequisites:The distance between the components of SCPR is neither too short not too long, and falls into the range 1.5–10 nm. Shorter distances result in very high frequency OOP ESE modulation, which is inaccessible for pulse EPR spectrometers. This frequency is determined by the magnetic dipolar interactions between the SCRP partners. For a short interspin distance (less than 1.5 nm), the exchange interaction between these spins can also contribute to OOP ESE modulation. Longer distances result in a very low frequency of OOP ESE modulation, so its variation cannot be detected before complete ESE decay due to the transversal relaxation time *T*_2_, which typically has a value of several microseconds.The spins of both components of the SCRP should be excited by microwave pulses. Practically, this means that the EPR spectrum of SCRP is not wider than a few tens of Gauss.The optimal magnetization turning angle for the first microwave pulse is different from π/2, unlike the normal in-phase ESE, but is close to π/4.The lifetime of the SCRP determined by its recombination is long enough for the ESE detection, i.e., at least several hundred nanoseconds.

The relatively low *j_SC_* value for P3HT/s-SWCNT is puzzling, taking into account the intense quenching of P3HT film photoluminescence upon s-SWCNT addition, reported in [6] and qualitatively reproduced in our experiments (Appendix A). On the one hand, since the photoluminescence quenching implies exciton splitting at the donor/acceptor interface, the initial charge photogeneration should be appreciable in the P3HT/s-SWCNT composite. On the other hand, very low *j_SC_* value implies that only a small fraction of the photogenerated charges becomes free and reaches the electrodes. This leads to the hypothesis that something happens with the photogenerated charges when they are coulombically bound and form the charge-transfer state. We used the OOP ESE technique to resolve this issue.

## 2. Results and Discussion

The dark in-phase echo-detected EPR signal of the P3HT/s-SWCNT composite was a structureless line centered at *g* = 2.0035 (Figure 1). Its intensity slightly increased upon continuous illumination. Very similar behavior was found for the pristine P3HT film at identical conditions, which is probably caused by stabilization of the photoinduced charges at defects of the P3HT. This allows the assignment of the dark EPR signal of the P3HT/s-SWCNT composite to the paramagnetic species in the P3HT, which survived the annealing at high vacuum. The transformation of the EPR spectrum of the P3HT/s-SWCNT composite under illumination drastically contrasts with that of the well-known photovoltaic composite P3HT/PC_60_BM [33]. For the latter, the light-induced EPR signal is much stronger than the dark signal, and the separate EPR lines for P3HT^+^ and PC_60_BM^−^ can be observed. No separate EPR signals for the species associated with the radicals localized at the s-SWCNT can be derived from the EPR spectra in Figure 1. This is in line with previous observations that the paramagnetic species associated with the SWCNTs are elusive [34].

The echo-detected EPR spectra of the P3HT/s-SWCNT composite obtained with the pulse laser excitation are shown in Figure 2. The spectra taken with synchronization of the laser and the microwave pulses with small delay after flash (DAF) were the superposition of two contributions [35]. The first contribution comes from the paramagnetic species generated by the laser flash immediately preceding the echo-forming two-pulse sequence, which can be either free charges or CTSs. They can carry non-equilibrium spin polarization. The second contribution comes from the stable paramagnetic defects and the paramagnetic species photoaccumulated due to large number of preceding laser flashes. These species are in thermal equilibrium. Since the lifetime of the CTS in this system (about 10 μs, see below) was much shorter than the shot repetition time in our experiments (1 ms), the spectrum recorded without synchronization of the laser and the microwave pulses contained only the species in thermal equilibrium, and the contribution of CTSs to this signal was very small. Therefore, the difference between the spectra recorded with and without synchronization of the laser and the microwave pulses contained only the signal of the species generated by the single laser pulse. The in-phase component of the “difference” spectrum was governed by net polarization of these species. In this respect, it is very similar to transient EPR spectrum [36,37], although the lines are slightly broader for the echo-detected EPR spectrum due to appreciable magnitude of the magnetic field of the microwave-pulsed *B*_1_. The out-of-phase component of the “difference” spectrum was solely governed by the CTS. As described above, it appeared due to correlation between the spin of the electron and the hole constituting the CTS. The out-of-phase component of the echo-detected spectrum obtained without synchronization of the laser and the microwave pulses (lower panel of Figure 2) contains only noise. The absence of out-of-phase ESE signal for this experiment is perfectly in line with our expectation. Since OOP ESE signal of the CTS P3HT^+^/s-SWCNT^−^ decayed within several microseconds after the laser flash (see below), the probability of the echo-detecting microwave pulse sequence falling within this interval was very low in the absence of the synchronization. Therefore, for the great majority of realizations of this pulse sequence, the contribution of the CTS P3HT^+^/s-SWCNT^−^ into the ESE signal was zero.

The in-phase echo-detected EPR spectrum in Figure 2 has an E/A pattern with the absorptive component centered at *g* = 2.003 and the weak emissive component expending to *g* = 2.008. The E/A pattern is typical for echo-detected EPR spectra of organic photovoltaic composites taken at low temperature with small DAF [38]. Such a pattern is generated by the spin evolution of the CTS during thermalization in the constant magnetic field of the EPR spectrometer, during which the spins of the charges constituting the CTS acquire opposite polarization [39,40,41]. Based on the comparison with the echo-detected EPR spectrum of the pristine P3HT, the absorptive peak at *g* = 2.003 should be assigned to the hole P3HT^+^. Accordingly, the broad emissive line should be assigned to the electron localized on the s-SWCNT. This is in line with the suggestion of J. Niklas et al. [34] that s-SWCNT^−^ has *g*-values up to 2.008.

The out-of-phase “difference” echo-detected EPR spectrum in Figure 2 was a single line, similar to that for other organic photovoltaic composites [42,43]. This observation implies that the mechanism of charge photogeneration in the P3HT/s-SWCNT composite is similar to that for common polymer/fullerene and polymer/non-fullerene acceptor composites. It also rules out the suggestion that no CTS is formed upon the photoexcitation of polymer/s-SWCNT composite [44].

The out-of-phase ESEEM traces for the P3HT/s-SWCNT composite (Figure 3) were also similar to those previously observed in organic donor/acceptor bulk heterojunction composites and can be modeled using the same approach [32]. The dependence of the out-of-phase ESE intensity *M_x_*(*τ*) on the interval *τ* between two echo-forming microwave pulses is determined by the interspin distance distribution function *G*(*R*):(1)$$M_{x}\left(\tau \right)=\exp\left(-2\tau /T_{2}\right)\int G\left(R\right)\iint \sin\left[\omega_{d}\left(R,\theta ,\phi \right)\tau \right]\sin\left(\theta \right)d\theta d\phi dR$$
where *T*_2_ is the transversal relaxation time for electron spins, *R* is the distance between the centers of spin density distributions of the electron and the hole, *θ* and *ϕ* are the polar and azimuthal angles, respectively, for the vector of the external magnetic field *B*_0_ in the reference frame of the principal axes of zero-field splitting (ZFS) tensor, determined by the dipolar interaction between the spin of the electron and the hole. The dipolar frequency *ω_d_* is determined by ZFS parameters *D* and *E*:(2)$$\omega_{d}=2\pi \left[\frac{2D\left(R\right)}{3}\left(1-3\cos^{2}\theta \right)+2E\left(R\right)\sin^{2}\theta \cos2\phi \right].$$

For point dipole approximation, which is assumed here because of the lack of the knowledge of the electron spin density distribution on s-SWCNT, *E*(*R*) = 0 and *D*(*R*) = 3/2 *γ*^2^/(*ħR*^3^), where *γ* is the free electron gyromagnetic ratio and *ħ* is the Planck constant.

Numerical simulations of out-of-phase ESEEM according to Equation (1) reproduced the experimental data for all three DAF values (Figure 3). The value *T*_2_ = 1.5 μs determined from the in-phase ESE decay was used for these simulations. As can be seen in Figure 3, no out-of-phase ESEEM can be observed in the pristine P3HT in the identical conditions, which testifies that all light-induced processes, except charge transfer from the P3HT to the s-SWCNT, can be neglected in OOP ESE modeling.

The distance distribution function between the components of the CTS P3HT^+^/s-SWCNT^−^ derived from the out-of-phase ESE simulation are shown in Figure 4. Since all these ESE traces were accumulated under identical conditions except for different DAF values, a quantitative comparison of the distance distribution density is possible. Again, the distance corresponding to the peak of the distribution density (several nanometers) and its tendency to increase with DAF is similar to those for CTS in other organic photovoltaics donor/acceptor composites [30,32]. However, the decay of *G*(*R*) with DAF increase for the CTS P3HT^+^/s-SWCNT^−^ is notably faster than that for CTS in well-known polymer/fullerene composite PCDTBT/PC_70_BM at cryogenic temperatures [45]. Presumably, it reflects faster geminate recombination of P3HT^+^/s-SWCNT^−^. Additionally, the most probable distance of the initial charge separation of 2.5 nm, which corresponds to the maximum of the distribution density for DAF = 200 ns in Figure 4, is smaller than the corresponding value for PCDTBT/PC_70_BM of 3.5 nm. Taking into account the exponential attenuation of the electron transfer rate with the distance between the recombining charges [46,47], the decrease of this distance by 1 nm for P3HT^+^/s-SWCNT^−^ can cause significant decrease in its lifetime compared to PCDTBT^+^/PC_70_BM^−^. Fast geminate recombination may be caused by strong electronic coupling between P3HT^+^ and s-SWCNT^−^ due to wrapping of the P3HT on the s-SWCNT [48,49,50,51]. This fast recombination also explains the low yield of photogenerated free charges in the P3HT/s-SWCNT composite, which could not be detected in CW EPR experiments with continuous light illumination due to low stationary concentration.

Although the preparation of the P3HT/s-SWCNT composites for our pulse EPR study and for the OPV device in [6] differ (different substrates and solvents), which causes a variation in the composite morphology, we do not expect that it affects significantly the CTS geometry. This is because the latter is determined mainly by the structure of the donor/acceptor interface. In the present case, this structure is determined by the interaction of the s-SWCNT with the P3HT chain wrapped on it. Since the s-SWCNT acts as a template for the P3HT, we suggest that the influence of the substrate on the aggregation of the s-SWCNT and the P3HT in films is moderate. The insensitivity of the CTS geometry to the substrate and the film deposition method was recently shown for the composite of a donor polymer with a non-fullerene acceptor by direct comparison of OOP ESEEM traces [43]. We also suggest that the structure of CTS at 30 K (the temperature of our ESE experiments) and at room temperature (the typical temperature of solar cell operation) are similar. This suggestion is based on the fact that, in both cases, the CTS formation upon thermalization of the electron and the hole proceeds with the release of excess energy (the difference between the exciton energy and the CTS energy) of several hundred meV, much larger than the thermal energy.

## 3. Materials and Methods

To prepare the P3HT/s-SWCNT composite for the EPR measurements, 98% purity IsoNanotube-S powder from NanoIntegris Inc., Boisbriand, QC, Canada, nominal diameter 1.2–1.7 nm) was first dispersed in 1-fluoronaphthalene at a concentration of 0.2 mg/mL using an ultrasonic bath with the dispersion time of 30 min. Then, regioregular P3HT (Ossila) was added to solution at a concentration 10 mg/mL. The resulting blended solution was sonicated for 30 min in an ultrasonic bath. Then, 0.1 mL of the solution was transferred to the quartz EPR tube. While sonicating in the ultrasonic bath, it was evacuated to 3 × 10^−5^ torr using a vacuum pump. This allowed the smooth evaporation of the solvent without intense bubbling and resulted in the P3HT/s-SWCNT film at the bottom of the EPR tube. The estimated thickness of the film is 1 μm. Then, the film was annealed at 140 °C for 10 min at 3 × 10^−5^ torr. The film of the pristine P3HT inside the EPR tube was prepared in a similar way, but without adding the s-SWCNT. The annealing step is needed to remove stable paramagnetic species from the sample, which are associated with oxygen-induced defects in PH3T [52]. We used 1-fluoronaphtalene instead of o-dichlorobenzene as the solvent for preparing EPR samples, because, according to our observations, o-dichlorobenzene tends to degrade upon prolonged sonication. These results provide information on a non-volatile precipitate after evaporation of o-dichlorobenzene and subjected to long-term sonication in the ultrasonic bath. No such precipitate was found for 1-fluoronaphtalene with a similar procedure. While the chemical structure of this precipitate is unclear at present, we suggest that it is the product of dechlorination of o-dichlorobenzene caused by ultrasonication followed by oligomerization or polymerization of phenyl rings. This is in line with the observed formation of terphenyls upon the photoreaction of o-dichlorobenzene solutions [53]. We suggest that the dechlorination of o-dichlorobenzene can be mechanically activated by ultrasound waves.

ESE measurements were performed on X-band ELEXSYS ESP-580E EPR spectrometer equipped with an ER 4118 X-MD-5 dielectric cavity inside an Oxford Instruments CF 935 cryostat. Temperature was kept at 80 K by the cold nitrogen gas flow. Lower temperatures were stabilized by the cold helium flow. To measure the in-phase light-induced echo-detected EPR signal, the sample inside the EPR resonator was continuously irradiated by diode laser with *λ* = 630 nm, output power 30 mW.

In the out-of-phase ESE experiments, the laser flashes from TECH-laser (Laser-export Co., Ltd., Moscow, Russia) with wavelength 527 nm, pulse duration of about 5 ns, and pulse repetition rate of 1 kHz were used to illuminate the sample. The energy of the flash reaching the sample was about 15 μJ. ESE signal was obtained using two-pulse microwave pulse sequence applied after a laser flash, namely, *Flash*—*DAF*—π/4—*τ*—π—*τ*—*echo* sequence, where *DAF* is the delay after laser flash, the π-pulse was of 24 ns nominal duration. The pre-saturating π/2-pulse was applied to suppress the in-phase ESE signal of the stable and long-living photoaccumulated paramagnetic species and to measure the out-of-phase ESE signals of CTS without distortion.

To prove that the s-SWCNTs that we use are indeed the individual nanotubes, they were spin-coated from the o-dichlorobenzene dispersion onto the silicon plate at 2000 rpm. Then, atomic force microscopy (AFM) images were collected using an NT-MDT Ntegra atomic force microscope (Appendix A). As expected, the features of a length of about 1 μm and height of 1.0–1.7 nm are clearly visible in the AFM image. They can be safely assigned to the SWCNTs.

## 4. Conclusions

Out-of-phase ESE spectroscopy allowed the identification of the main reason for low yield of free charge photogeneration in P3HT/s-SWCNT OPV devices: fast geminate recombination of the P3HT^+^/s-SWCNT^−^ charge-transfer state. This brings into question the successful application of single-walled SWCNTs as the acceptor component of the active layer of OPV devices. On the other hand, they can be successfully applied for modifying the morphology of polymer/fullerene active layer [54,55]. In view of the present results, this seems to be the more promising direction of use for SWCNT additives to donor/acceptor OPV composites. 

## Figures and Tables

**Figure 1 ijms-24-04098-f001:**
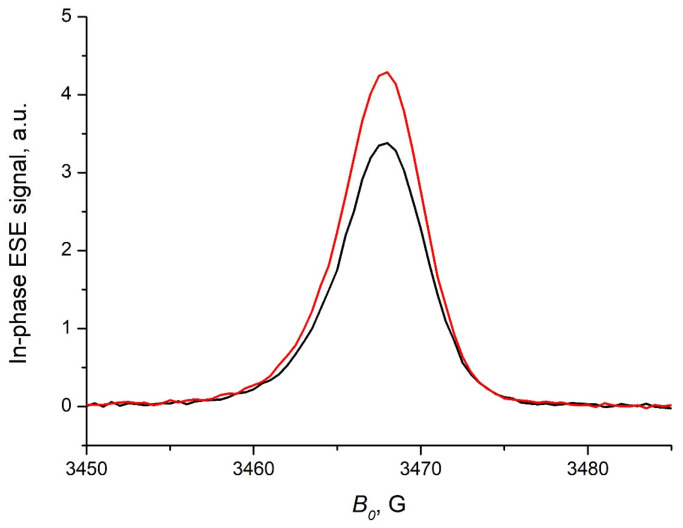
The echo-detected EPR spectra of the P3HT/s-SWCNT composite film obtained at 80 K in the dark (black line) and under continuous illumination (red line).

**Figure 2 ijms-24-04098-f002:**
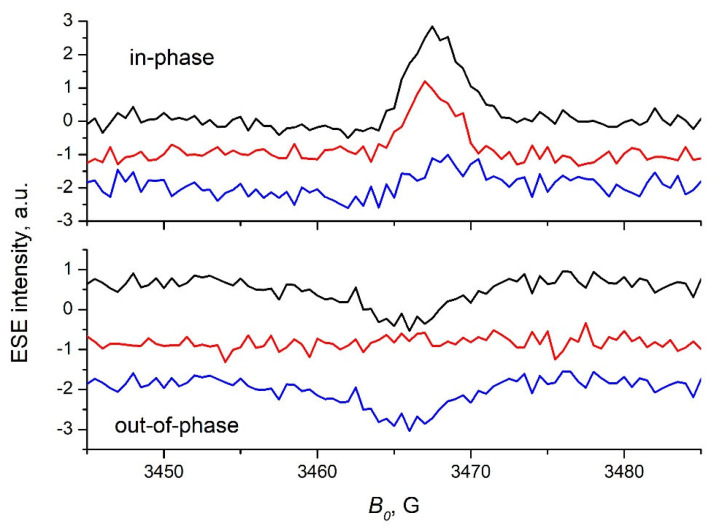
The echo-detected EPR spectra of the P3HT/s-SWCNT composite film taken at 30 K under pulsed laser illumination with delay after flash 200 ns (black lines), without synchronization of laser and microwave pulses (red lines) and their difference (blue lines). Upper panel—the in-phase component of ESE; lower panel—the out-of-phase component of ESE.

**Figure 3 ijms-24-04098-f003:**
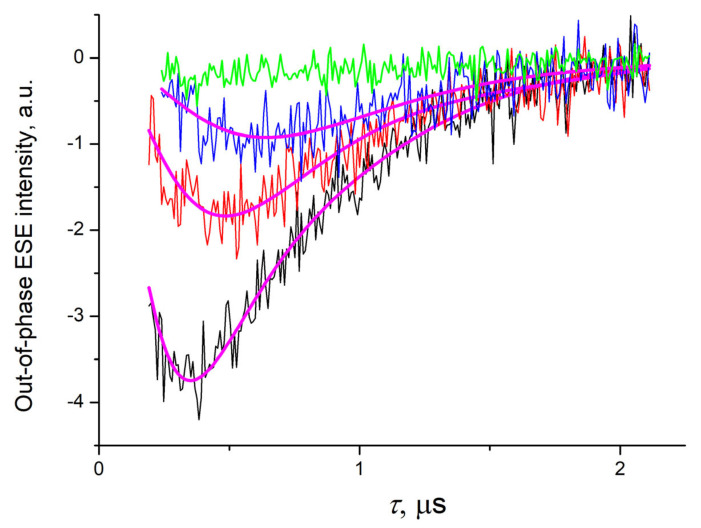
The out-of-phase ESEEM traces for the P3HT/s-SWCNT composite film taken at 30 K with DAF = 200 ns (black line), DAF = 10 μs (red line), and DAF = 30 μs (blue line) with their numerical simulation (thick magenta lines). Green line shows the out-of-phase ESEEM trace for the pristine P3HT film taken at 30 K with DAF = 200 ns.

**Figure 4 ijms-24-04098-f004:**
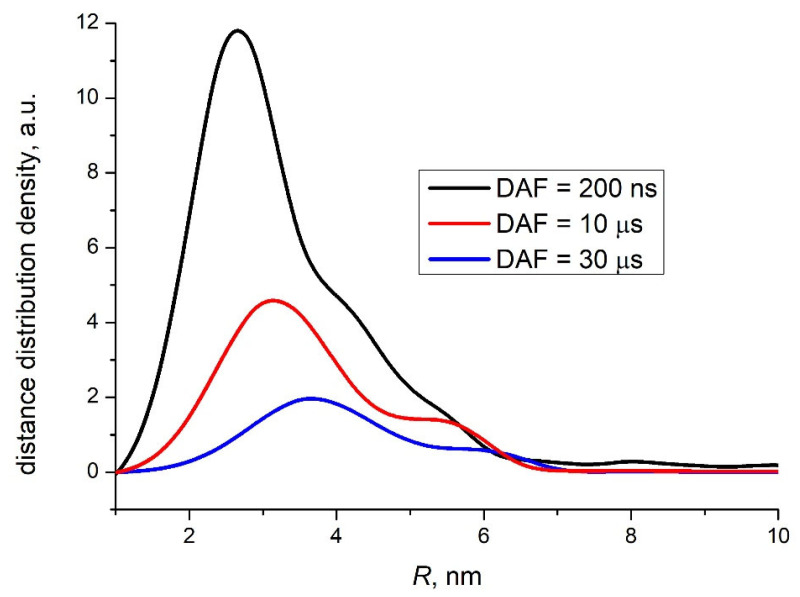
The interspin distance distribution density for the CTS P3HT^+^/s-SWCNT^−^ obtained from numerical simulation of the out-of-phase ESEEM traces for different DAF values: black line—200 ns, red line—10 μs, blue line—30 μs.

## Data Availability

The data presented in this study are available on request from the corresponding author.

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
