# Peer review of "Fast Recombination of Charge-Transfer State in Organic Photovoltaic Composite of P3HT and Semiconducting Carbon Nanotubes Is the Reason for Its Poor Photovoltaic Performance"

_ijms, 2023, doi:10.3390/ijms24044098_

Round 1
Reviewer 1 Report
The manuscript studies the reason for unsuccessful reproduction of the previous results: Ren, S. et al., Nano Lett. 2011, 11, 5316–5321 using semiconducting single-walled carbon nanotubes (s-SWCNT) in OPVs. The authors made an argument that the out-of-phase ESE spectroscopy allowed to identify the main reason for the fast geminate recombination of P3HT/s-SWCNT charge-transfer state. The efforts on understanding geminate recombination rate of donor-acceptor system by ESE spectroscopy can be interesting to the readers, however, there are some major and minor things to be addressed before publishing.
1. The preparation of P3HT/s-SWCNT samples for ESE spectroscopy is essentially different from the original P3HT/s-SWCNT photoactive layer (PAL) in OPV device. Especially the blend morphology of donor-acceptor pair will be completely different since the coating media (solvent) and the surface energy of the underlying substrate changed even though the annealing procedure was the same.
The difference can be seen from the huge different thicknesses of the PALs (60 nm vs 1000 nm) which presumably much affected the donor-acceptor blend morphology. Additionally, Figure 2 in Ref. 6 shows AFM image of ~10 nm of the P3HT/s-SWCNT morphology while the manuscript only show Figure S1 with dispersed s-SWCNTs which could not prove that the blend morphology of the layer will be the same. Can the authors explain more on this?
2. The authors emphasized "fast recombination of charge transfer state" even in the title of the manuscript. The "fast" terminology seems to be vague and inaccurate. Could the authors show the control data or comparison data of well-known system to demonstrate the relative recombination rate of this particular system?
3. Minor things: a) page 4, line 176-181. A lot of typos can be found. b) The use of comma, as the separation of decimal point should be consistent with the international standard.
Author Response
The authors are thankful to the reviewers for their instructive comments. Most of their suggestion is implemented into the revised version of the manuscript. The point-by-point answers on the questions raised by the reviewers are below.
Reviewer 1
Question of the reviewer
- The preparation of P3HT/s-SWCNT samples for ESE spectroscopy is essentially different from the original P3HT/s-SWCNT photoactive layer (PAL) in OPV device. Especially the blend morphology of donor-acceptor pair will be completely different since the coating media (solvent) and the surface energy of the underlying substrate changed even though the annealing procedure was the same.
The difference can be seen from the huge different thicknesses of the PALs (60 nm vs 1000 nm) which presumably much affected the donor-acceptor blend morphology. Additionally, Figure 2 in Ref. 6 shows AFM image of ~10 nm of the P3HT/s-SWCNT morphology while the manuscript only show Figure S1 with dispersed s-SWCNTs which could not prove that the blend morphology of the layer will be the same. Can the authors explain more on this?
Answer of the authors
Indeed, the morphology of the active layer is highly important for OPV device operation and it is affected by preparation of donor/acceptor composite, in particular, by the substrate, the processing solvent and post-annealing. However, the composite morphology is not so important for the structure of CTS, as we explained in detail at the end of Results and Discussion section:
“Although the preparation of P3HT/s-SWCNT composites for pulse EPR study and for OPV device differ (different substrates and solvents), which causes the variation of composite morphology, we do not expect that it affects significantly the CTS geometry. This is because the CTS geometry is determined mainly by the structure of donor/acceptor interface. In the present case this structure is determined by the interaction of the s-SWCNT with P3HT chain wrapped on it. Since s-SWCNT acts as a template for P3HT, we suggest that the influence of the substrate on the aggregation of s-SWCNT and P3HT in films is moderate. The insensitivity of the CTS geometry to the substrate and film deposition method was recently shown for the composite of a donor polymer with a non-fullerene acceptor by direct comparison of OOP ESEEM traces.46”
Concerning AMF images of P3HT/s-SWCNT composite we (Figure 2b in Ref. 6) we should note that AFM is not a particularly informative method to study this system. Moreover, the interpretation of this image in Ref. 6 is probably misleading. First, the length of the “worms” in this image (50 – 100 nm) is too short to be caused by carbon nanotubes with typical length of 1 micrometer. Second, such a morphology is often obtained for pristine P3HT films, without CNT adding [Brinkmann, M. Structure and Morphology Control in Thin Films of Regioregular Poly(3-Hexylthiophene). J. Polym. Sci. Part B Polym. Phys. 2011, 49 (17), 1218–1233. https://doi.org/10.1002/polb.22310]. The typical width of the “worms” (10 nm) can be determined by crystalline lamellae in pristine P3HT films [Brinkmann, M.; Rannou, P. Macromolecules 2009, 42, 1125].
Question of the reviewer
- The authors emphasized "fast recombination of charge transfer state" even in the title of the manuscript. The "fast" terminology seems to be vague and inaccurate. Could the authors show the control data or comparison data of well-known system to demonstrate the relative recombination rate of this particular system?
Answer of the authors
We used the data on CTS recombination in bulk heterojunction composite PCDTBT/PC70BM as a control. This system is well-known because it demonstrated record-high OPV efficiency in 2009 [S. H. Park et al., “Bulk heterojunction solar cells with internal quantum efficiency approaching 100%,” Nat. Photonics 3, 297–302 (2009)] and was extensively studied later. The detailed comparison is given at Results and Discussion section (before Fig. 4):
“…the decay of G(R) with DAF increase for CTS P3HT+/s-SWCNT ̵ is notably faster than that for CTS well-known polymer/fullerene composite PCDTBT/PC70BM at cryogenic temperature.48 Presumably, it reflects faster geminate recombination of CTS P3HT+/s-SWCNT ̵. Also, the most probable distance of initial charge separation 2.5 nm, which corresponds to the maximum of distribution density for DAF = 200 ns in Fig. 4, is smaller than the corresponding value for PCDTBT/PC70BM – 3.5 nm. Taking into account exponential attenuation of the electron transfer rate with the distance between the recombining charges,49,50 decrease of this distance by 1 nm for P3HT+/s-SWCNT ̵ can cause significant decrease of its lifetime, compared to CTS PCDTBT+/PC70BM ̵. Fast geminate recombination may be caused by strong electronic coupling between P3HT+ and s-SWCNT ̵ due to wrapping of P3HT on s-SWCNT.51–54 This fast recombination also explains the low yield of photogenerated free charges. In P3HT/s-SWCNT composite, which could not be detected in CW EPR experiments with continuous light illumination due to low stationary concentration.”
Question of the reviewer
- Minor things: a) page 4, line 176-181. A lot of typos can be found. b) The use of comma, as the separation of decimal point should be consistent with the international standard.
Answer of the authors
The typos are corrected.
Reviewer 2 Report
This submitted manuscript was an interesting read, and there was some insight that I feel is publishable in this journal... Therefore, I would suggest that the manuscript be accepted after (many) minor revisions. Below are some broad questions, and in the attached PDF, I have added some other questions in red.
- If I am reading between the lines, I feel that the authors might be suggesting that "Ref 6 is not reproducible... and here is why!..." The authors do not outright say this, but if they believe it to be true, then they probably should. If they believe that it is only "not reproducible in our hands..." then they should use the conclusion section to speculate their results are so different from Ref 6. What else could have went wrong in the solar cell fabrication between your cells and theirs? Perhaps provide a critique of your own cell fabrication.
- There is lots of good detail about a technique that is not common (and I used EPR throughout my PhD), and therefore it would be good to include a brief tutorial on pg 2. Also... It would be good to clarify why a signal at 30 K is relevant in an operational room temperature solar cell.
- Finally the grammar and spelling needs major work. The highlights in the attached document represent all of the grammar/spelling/formatting errors that i found... and that will need to be cleaned up for the final draft.

Author Response
Question of the reviewer
If I am reading between the lines, I feel that the authors might be suggesting that "Ref 6 is not reproducible... and here is why!..." The authors do not outright say this, but if they believe it to be true, then they probably should. If they believe that it is only "not reproducible in our hands..." then they should use the conclusion section to speculate their results are so different from Ref 6. What else could have went wrong in the solar cell fabrication between your cells and theirs? Perhaps provide a critique of your own cell fabrication.
Answer of the authors
Since Ref. 6 contains numerous inconsistencies we believe that its main results are irreproducible in principle. In particular, extremely high VOC value for OPV devices with P3HT/s-SWCNT composite as the active layer violates basic rules of semiconductor physics. As we stated in the Conclusion section, the most probable reason for this is some admixture to the P3HT/s-SWCNT composite:
“The origin of dramatic discrepancy between our results and those of Ref. 6 is unclear at present. However, we note that photoluminescence spectra of P3HT nanofilaments (Fig. 3b of Ref. 6) with maximum at 540 nm are inconsistent with literature data of photoluminescence of P3HT films, which demonstrate maximum in the range 630 – 730 nm depending on the processing solvent.59–63 This implies the presence of some other substance in P3HT films studied in Ref. 6, which could cause the discrepancy. The presence of additional species also in P3HT/s-SWCNT composite and their participation in photoelectric conversion can naturally explain very high VOC value reported in Ref. 6.”
Question of the reviewer
- There is lots of good detail about a technique that is not common (and I used EPR throughout my PhD), and therefore it would be good to include a brief tutorial on pg 2.
Answer of the authors
Very brief tutorial on out-of-phase ESE spectroscopy is added to the end of the Introduction section:
“For successful measurement of the distance between the components of SCPR using OOP ESE technique the following prerequisites are required:
- The distance between the components of SCPR is neither too short not too long and falls into the range 1.5 – 10 nm. Shorter distances result in very high frequency of OOP ESE modulation, which is inaccessible for pulse EPR spectrometers. Longer distances result in very low frequency of OOP ESE modulation, so its variation can not be detected before complete ESE decay with transversal relaxation time T2 in order of several microseconds.
- Spins of both components of SCRP should be excited by microwave pulses.
- The optimal magnetization turning angle for the first microwave pulse is different from π/2 in contrast to normal in-phase ESE, but is close to π/4.
- Lifetime of SCRP is long enough for ESE detection, i. e. at least several hundred nanoseconds.”
Question of the reviewer
It would be good to clarify why a signal at 30 K is relevant in an operational room temperature solar cell.
Answer of the authors
The explanation is added to the end of Results and Discussion section:
“We also suggest that the structure of CTS at 30 K (the temperature of our ESE experiments) and at room temperature (typical temperature of solar cell operation) are similar. This suggestion is based on the fact that in both cases CTS formation upon thermalization of the electron and the hole proceeds with release of excess energy (the difference between the exciton energy and the CTS energy) of several hundred meV – much larger than the thermal energy. “
Question of the reviewer
Finally the grammar and spelling needs major work. The highlights in the attached document represent all of the grammar/spelling/formatting errors that i found... and that will need to be cleaned up for the final draft.
Answer of the authors
The authors appreciate the pointing to so many grammar/spelling/formatting errors. These errors are corrected.
Round 2
Reviewer 1 Report
The authors fulfilled all of my questions. Therefore, this reviewer suggest the manuscript to be published.
Author Response
The authors thank the reviewer again for valuable comments.